# Synthesis and Characterization of Amorphous Iron Oxide Nanoparticles by the Sonochemical Method and Their Application for the Remediation of Heavy Metals from Wastewater

**DOI:** 10.3390/nano10081551

**Published:** 2020-08-07

**Authors:** Virendra Kumar Yadav, Daoud Ali, Samreen Heena Khan, Govindhan Gnanamoorthy, Nisha Choudhary, Krishna Kumar Yadav, Van Nam Thai, Seik Altaf Hussain, Salim Manhrdas

**Affiliations:** 1School of Lifesciences, Jaipur National University, Jaipur, Rajasthan 302017, India; yadava94@gmail.com; 2Department of Zoology, College of Science, King Saud University, Riyadh 11451, Saudi Arabia; huasainaltaf2@gmail.com (S.A.H.); Salem1752@outlook.com (S.M.); 3School of Nanosciences, Central University of Gujarat, Gandhinagar, Gujarat 382030, India; samreen.heena.khan@gmail.com (S.H.K.); nishanaseer03@gmail.com (N.C.); 4Department of inorganic chemistry, University of Madras, Guindy Campus, Chennai T.N. 600025, India; gnanadrdo@gmail.com; 5Institute of Environment and Development Studies, Bundelkhand University, Kanpur Road, Jhansi 284128, India; envirokrishna@gmail.com; 6Ho Chi Minh City University of Technology (HUTECH) 475A, Dien Bien Phu, Ward 25, Binh Thanh District, Ho Chi Minh City 700000, Vietnam

**Keywords:** sonochemical, fly ash, acoustic cavitation, iron oxide nanoparticlXes, magnetite

## Abstract

Nanoparticles have gained huge attention in the last decade due to their applications in electronics, medicine, and environmental clean-up. Iron oxide nanoparticles (IONPs) are widely used for the wastewater treatment due to their recyclable nature and easy manipulation by an external magnetic field. Here, in the present research work, iron oxide nanoparticles were synthesized by the sonochemical method by using precursors of ferrous sulfate and ferric chloride at 70 °C for one hour in an ultrasonicator. The synthesized iron oxide nanoparticles were characterized by diffraction light scattering (DLS), Fourier transform infrared spectroscopy (FTIR), Raman spectroscopy, X-ray diffraction (XRD), field-emission scanning electron microscopy (FESEM), electron diffraction spectroscopy (EDS), high-resolution transmission electron microscopy (HRTEM) and vibrating sample magnetometer (VSM). The FTIR analysis exhibits characteristic absorption bands of IONPs at 400–800 cm^−1^, while the Raman spectra showed three characteristic bands at 273, 675, and 1379 cm^−1^ for the synthesized IONPs. The XRD data revealed three major intensity peaks at two theta, 33°, 35°, and 64° which indicated the presence of maghemite and magnetite phase. The size of the spherical shaped IONPs was varying from 9–70 nm with an average size of 38.9 nm while the size of cuboidal shaped particle size was in microns. The purity of the synthesized IONPs was confirmed by the EDS attached to the FESEM, which clearly show sharp peaks for Fe and O, while the magnetic behavior of the IONPs was confirmed by the VSM measurement and the magnetization was 2.43 emu/g. The batch adsorption study of lead (Pb) and chromium (Cr) from 20% fly ash aqueous solutions was carried out by using 0.6 mg/100 mL IONPs, which exhibited maximum removal efficiency i.e., 97.96% and 82.8% for Pb^2+^ and Cr ions, respectively. The fly ash are being used in making cements, tiles, bricks, bio fertilizers etc., where the presence of fly ash is undesired property which has to be either removed or will be brought up to the value of acceptable level in the fly ash. Therefore, the synthesized IONPs, can be applied in the elimination of heavy metals and other undesired elements from fly ash with a short period of time. Moreover, the IONPs that have been used as a nanoadsorbent can be recovered from the reaction mixture by applying an external magnetic field that can be recycled and reused. Therefore, this study can be effective in all the fly ash-based industries for elimination of the undesired elements, while recyclability and reusable nature of IONPs will make the whole adsorption or elimination process much economical.

## 1. Introduction

Heavy metals are one of the major inorganic pollutants arising from the iron and steel industries, electroplating industries, and battery industries [1]. These heavy metals are non-degradable, and persistent in nature, due to which they can only be transformed from a toxic form to a non-toxic form [2]. Such toxic forms of heavy metals may persist for a longer time in our environment and once they come contact with land, water and soil this may lead to potential threat to livings beings [3]. Although in the environment there are various sources of discharge of heavy metals, here we are trying to emphasize a globally hazardous material i.e., fly ash. Fly ash is a byproduct of thermal power plants (TPPs), produced during combustion of pulverized coal during generation of electricity [4]. Fly ash is considered a hazardous material due to the presence of Pb, Cd, Cr, Cu, Zn, Ni, Hg, Mo, As, and Co as major heavy metals in it [5]. As a result, there is limited acceptance of fly ash as a raw material for manufacturing of value added products. In India, more than one million tonnes [MTs] of fly ash is generated every year [4] and for 2019–2020 it was 1.25 MTs. A major part of this fly ash (about 50–60%) is generally dumped near the thermal power plants in the fly ash ponds [6]. During the rainy season, these fly ash ponds comes in contact with rain water and results into leaching of heavy metals, to the nearby water bodies, ground water or agricultural land [7]. Subsequently, these heavy metals bio-accumulate in the plants and this way enter into the food chain. The lower animals feed on such contaminated plants and get affected. When such animals are consumed by higher animals (or human beings) in food chain [8] then, there will be bio magnification of heavy metals [9]. This will cause severe diseases in human beings like neurological disorders, vomiting, diarrhea, numbness, and in chronic cases may lead to death [10]. However, fly ash are loaded with numerous heavy metals but here we emphasise Pb and Cr, as diseases associated with both of these heavy metals have increased in recent years due to contaminated water bodies, groundwater pollution, etc. Chromium (Cr), exists in three different forms i.e., Cr^2+^, Cr^6^, and Cr^6+^, out of which Cr^6+^ is carcinogenic [11]. From the previous, fly ash based heavy metal leaching experiment, it was found that the Cr is mainly present as Cr^6+^ ions in the aqueous leachate. If Cr^6+^ is transformed into other forms of Cr, then it will be non-carcinogenic and non-toxic.

The presence of heavy metals in fly ash could be the darker side of the fly ash but presence of valuable minerals is the brighter side. Due to this, it has gained tremendous applications in the field of civil engineering, bricks, tiles, pavement blocks, cements [12], kitchen panels, hydrometallurgy [13] and biofertilizer [14]. Fly ash has a surplus amount of macro and micro nutrients required by plants. Moreover, 70–90% of the fly ash fraction is made up of silica, alumina and ferrous, which can be recovered from fly ash by means of chemical approaches [15]. But, the presence of these heavy metals in fly ash hinders their application in civil engineering, metallurgy and as bio fertilizers. This could be due to the leaching of heavy metals from fly ash-based finished products. This may further pose a potential threat to consumers due to the leaching of heavy metals. So, prior to the utilization of fly ash as a raw material in all the aforementioned industries it is utmost important to either eliminate the heavy metals, i.e., Pb and Cr or bring down the concentration of both the heavy metals up to acceptable range.

Therefore, one of the most promising technology for the elimination or remediation of Pb and Cr, heavy metals from fly ash aqueous solution is nanotechnology [16,17]. Due to the unique physical and chemical properties of nanoparticles at nanoscale [18], they are widely used for wastewater treatment [19] and environmental clean-up [20]. However, numerous metal oxide nanoparticles like alumina (Al_2_O_3_) [16,21] zinc oxide (ZnO) [22], titanium dioxide (TiO_2_) [23] and iron oxide nanoparticles (IONPs) [19,24] have been shown to have promising potential as a nanoadsorbent for environmental clean-up. Out of all the above mentioned nanoparticles, IONPs in their amorphous form are most preferred candidate for heavy metal remediation. This is because of their high surface area to volume ratio and high surface energies [25]. Besides, these IONPs are also recyclable, low cost, and could be easily manipulation by an external magnetic field [26]. The amorphous IONPs could be synthesized by physical, chemical and microbial routes. Of these three methods, the chemical method is most preferred due to their easiness, less time taking and requirement of less expensive instrument. By means of chemical routes, IONPs have been successfully synthesized by co-precipitation method [19], solvothermal method [27] microemulsion method [28], sol-gel method [29], chemical vapor deposition method [30] and sonochemical method [23,31]. However, when it comes to the requirement of an amorphous nanoparticles with a better control of the morphology then sonochemical is the preferred method [32]. This is because the reaction parameters such as temperature, sonication time, and power of ultrasonication plays an important role in the morphology of the final products. Sonochemistry is a novel and simple route for [33] which applies acoustic cavitation for the synthesis of the nanoparticles, where the particles are broken down into smaller sizes with the bubbles [34]. The ultrasound is preferred for the synthesis of nanoparticles not only due to its simplicity and diverse applicability [35], but also due mixing of the constituent ions at the atomic level. Due to the mixing at atomic level, there is formation of amorphous phase [36] of nanoparticles which can be later transformed into the crystalline phase by simple annealing or calcination at a relatively lower temperature [37,38]. The ultrasonic irradiation causes cavitation in an aqueous medium, which can generate a temperature of around 5000 °C and a pressure of over 1800 kPa, which enables many unusual chemical reactions to occur [39]. The cavitation is marked by the sequential events i.e., formation, growth and collapse of the microbubbles [40].

Numerous investigators have reported the sonochemical synthesis of IONPs either functionalized or naked and applied them in the field of medicine and environmental cleanup. One such work was carried out by Vijaya et al. and synthesized transition metal oxide nanoparticles including Fe_3_O_4_, by using iron acetate as a precursor material. The IONPs was synthesized by using 10% water-dimethylformamide [DMF], irradiated with a high-intensity ultrasonic horn (Ti-horn, 20 kHz, 100 W/cm^2^) under 1.5 atm of argon at room temperature (RT) for three hours. The size of the synthesized IONPs was spherical in shape whose size was in nano range which was confirmed by transmission electron microscopy (TEM). The reaction involves use of argon, in addition to long reaction time of three hours which is the disadvantage of the reaction [41]. Hassanjani-Roshan et al. synthesized IONPs (Fe_2_O_3_) by sonochemical method, and analyzed by TEM, X-ray diffraction (XRD), and thermo gravimetric and differential thermal analyses and found that the size of IONPs was smaller than 19 nm and exhibit superparamagnetism. They synthesized IONPs at (30 °C, 60 °C, and 80 °C), and the sample was calcinated at 500 °C, for 1 h to obtain crystalline IONPs [42].

Therefore, here we tried to overcome the drawback of both the above experiments, as former experiment have specific synthesis conditions and time consuming step, while the latter experiment performs calcination which transforms the amorphous IONPS into crystalline. As, it is reported that the amorphous IONPs have wider applications in industry in comparison to the crystalline IONPs. So, here we synthesized IONPs by sonochemical method at RT within one hour only and the as synthesized powder was used for the elimination of Pb and Cr heavy metal from the fly ash solution.

In the present research work, we have suggested a method for the elimination of Pb and Cr heavy metal ions from the fly ash aqueous solutions using amorphous IONPs synthesized by the sonochemical method. The sonochemical synthesis was undertaken at 70 °C for one hours without using any argon or other gas environment. The confirmation of the formation and purity of the sample was undertaken by diffraction light scattering (DLS), Fourier transform infrared spectroscopy (FTIR), Raman spectroscopy, XRD, field-emission scanning electron microscopy (FESEM), electron diffraction spectroscopy (EDS), high-resolution transmission electron microscopy (HRTEM) and vibrating sample magnetometer (VSM). Further, the synthesized IONPs were used for the remediation of Pb^2+^ and Cr ions heavy metals from the 20% fly ash aqueous solutions, at room temperature, at a fixed dose of IONPs. The remediation was studied w.r.t to the contact time, where the concentration of Pb and Cr was analyzed by the inductively coupled plasma-optical emission spectroscopy (ICP-OES). Here one of our objective is to eliminate both Pb^2+^ and Cr ions heavy metals from fly ash aqueous solutions. However, another objective was to assess the adsorption potential of amorphous IONPs. Such practice will help in categorizing the fly ash from hazardous material to non-hazardous material category [43].

## 2. Materials and Methods

### 2.1. Materials

Iron (III) chloride (FeCl_3._) anhydrous (Merck, Germany), Sodium hydroxide (NaOH) pellets (Merck, Germany), Iron sulfate heptahydrate (FeSO_4_.7H_2_O) (Himedia, India) and Ethanol (Xilong Scientific Co., Ltd, Shantou, Guangdong, China) were all of analytical grade and used without purification. Round bottom flask 100 mL, 100–200 mL glass beaker (2–3 no’s), Petri plates, fly ash collected from Gandhinagar Thermal power plant, (Gandhinagar, Gujarat, India), Milli-Q water (Milli-Q, Type 1 Ultrapure Systems, Merck, Millipore. Germany) and magnet (procured from A-Z magnet, Chandni Chowk, New Delhi, India).

### 2.2. Synthesis of Iron Oxide Nanoparticles (IONPs)

An aqueous solution of 0.2 mmol FeCl_3_ anhydrous was prepared by dissolving in 40 mL of Milli-Q water while 0.1 mmol of FeSO_4_.7H_2_O were prepared by dissolving in 40 mL of Milli-Q water in order to keep the ratio molar ratio 2:1 [44]. The aqueous solutions of both the salts were added to a 100 mL round bottom flask which was later kept in an ultrasonicator (Sonar, 40 kHz), along with heating at 60–70 °C. An aqueous solution of 4M NaOH was prepared in the milli-Q water which was continuously added to the mixture drop wise until the pH rose to 11–13. Once the pH of the mixture raised to 11–13, NaOH addition was stopped and there was the formation of a black precipitate at the bottom of the flask. Further, the flask was closed by a cork and the reaction was allowed to continue for one hour in the sonicator at the above-mentioned parameters. Once the reaction was over, the mixture was allowed to cool at room temperature (RT) followed by centrifugation at 6000 rpm for 5 min. The black precipitate was retained while the supernatant was discarded. Figure 1A is showing mixture of IONPs before drying and the Figure 1B is showing mixture of IONPs responding towards external magnetic field. Further, the obtained precipitate was washed 2–3 times each with milli-Q water and ethanol. Finally, the precipitate was dried in an oven at 40 °C for overnight to obtain the dried powder. Shrivastav et al. 2012 and Yadav and Fulekar 2017, also reported the synthesis of IONPs by sonochemical method, by using a similar protocol.

### 2.3. Fly Ash Collection And Preparation of 20% Fly Ash Aqueous Solution

About 1 kg of fly ash was collected directly from the electrostatic precipitator, from Gandhinagar Thermal power plant, Gujarat, India, in a sterilized plastic silo which was dried at 110 °C in a hot air oven for 24 h. Further, 200 g of fly ash was mixed with one liter of milli-Q water to obtain a 20% fly ash aqueous solution. The containers containing the mixture was sealed and shaken in a horizontal shaker at 150 rpm at RT (25–26 °C) [19,45] for 24 h for proper mixing. Furthermore, the mixture was allowed to stand for 12 h before the leachate was collected [46]. Finally, the leachate was collected from the mixture by filtering it through a Whatman filter paper no. 42. Few drops of analytical grade nitric acid were added to the leachate and the containers were kept in a refrigerator at 4 °C to prevent any additional chemical reaction.

### 2.4. Batch Adsorption Study of Pb and Cr Ions

Batch adsorption studies were performed on the basis of prior conducted adsorption experiments to optimize the concentrations and reaction conditions. For final adsorption study, 0.6 mg of IONPs were added in 100 mL of 20% fly ash aqueous solution in a 250 mL conical flask and placed in an incubator shaker at 30 °C and at 150 rpm. Here the dose of IONPs, temperature, pH, and rpm of the adsorption reaction was kept fixed while the contact time was variable. The duration of the interaction of the particle was for 24 h, where the sample was supposed to be collected at the interval of 1 h, 2 h, 4 h, 6 h, 8 h, 12 h, 16 h, 18 h and 24 h. Firstly, an aliquot was taken from the stock at 0 h for the estimation of concentration of Pb^2+^ and Cr ions in the sample. Further, an aliquot around (5–10 mL) was collected and analyzed by the ICP-OES at the given time interval for the detection of the concentration of Pb^2+^ and Cr ions in the sample.

### 2.5. Characterization of IONPs

An aqueous solution of IONPs was prepared by adding a few (~2) mg of powder into the milli Q water, which was dispersed for 10 min in an ultrasonicator device (Sonar 40 kHz). Furthermore, an aliquot (1 mL) was taken from the dispersed solution for DLS analysis. The DLS measurement was done at RT, by using (Microtrac Zetatrac, U2771, DLS XE-70, Park System equipment) to obtain the average particle size and polydispersity index (PDI). Further, a drop of the dispersed sample was cast on a carbon-coated copper grid for the image analysis by HRTEM by using [FEI Model Tecnai G2 S Twin (200 kV)]. FTIR (Perkin-Elmer One Spectrum 6500, USA), FESEM-EDS (SEM Carl Zeiss Germany, Model no. NOVA NANOSEM-450), and powder XRD (Bruker, D8 Advance) analyses were performed by using solid powder samples. For FTIR, IONPs were mixed with potassium bromide (KBr) powder (FTIR grade), mixed finely, in a mortar pestle, and a pellet was prepared by using a mechanical press machine. The pellet was examined against a blank KBr pellet in the range of 400–3600 cm^−1^ at one nm resolution in the transmittance mode. For Raman analysis, the powder sample of IONPs was placed on the glass slide, and measurement was done in the range of 200–4000 cm^−1^ with an excitation laser line of 632.8 cm^−1^ from a He-Ne laser, by using (Witec Germany—In Via Raman). For the surface morphology examination by FESEM, a pinch of IONPs sample was placed on the surface of carbon tape, and fixed on an aluminum tape. The sample along with aluminum stub was placed in a gold sputter machine for 10 min for Au sputtering. The elemental composition of the IONPs was detected by the attached elemental analyzer (EDS, Oxford). For, EDS analysis, a region was selected and data was acquired by focusing on a particular area While for powder XRD, the sample was analyzed in the range of two theta, 5–70°, and data was accorded. The magnetic behavior of the sample was done by a VSM [Quantum Design PPMS Model 6000 magnetometer]. For analysis, a pellet was prepared which was coated with the Teflon tape and prior to the analysis sample was placed in the quartz probe. Saturation magnetization was obtained from magnetization curves at room temperature by applying magnetic fields from −2500 to +2500 T. The heavy metal analysis of fly ash digested sample and 20% aqueous solutions was analyzed by the using ICP-OES. It helps in the detection of trace elements in the liquid sample. All the samples were filtered and the analysis was undertaken by SPECTRO ARCOS, Analytical instruments GmbH, Germany simultaneous inductively coupled plasma (ICP) spectrometer.

## 3. Results and Discussion

The formation of IONPs by the sonochemical method is due to acoustic cavitation, which involves three basic sequential steps; formation of bubbles followed by growth and implosive collapse that result in high pressure as well as temperature followed by the high cooling rate [32]. The chemical reactions which are involved in the sonochemical method are reported to be driven by intense ultrasonic waves that are strong enough to produce cavitation are oxidation, reduction, dissolution, decomposition, and hydrolysis [47]. Initially, there is a formation of a minute bubble that gradually grows in size due to the fusion of several small particles and once the size of the bubbles reaches to its maximum i.e., threshold than the bubble could not sustain the pressure difference, and consequently, the bubble implosively collapses and generates a localized hotspot through adiabatic compression or shock wave formation within the gas phase of the collapsing bubble [48]. Again, there will be the formation of the bubble, followed by growth, and finally the implosive collapse of the bubble. All these steps keep on repeating which ensures not only their amorphous nature but also their size in the nano range. In the sonochemical method the particles which are nucleated could not attain the proper crystalline structure, due to which it generally produces amorphous nanomaterials [49]. The pH solution of the mixture during synthesis of the IONPs was 11–13, while the initial pH of the 20% fly ash aqueous solution was 7.6 while after the completion of the batch experiment the pH raised to 8.5–9. This was because the fly ash have Ca and other alkali metals which form hydroxides with water and increases the pH of the solution. The increase in the pH was at steady state so it was not affecting the rate of removal of both the heavy metals from the fly ash solution.

### 3.1. Fourier Transform Infrared (FTIR) and Raman Spectroscopy of Synthesized IONPs

FTIR is one of the analytical techniques that are used to reveal the functional groups present in the sample and also to determine the microcrystalline nature of the particles. In Figure 2, the FTIR spectra reveal the three characteristic peaks at 552 cm^−1^, 645 cm^−1^, and 990 cm^−1^ which are attributed to the Fe-O bond [50]. The band at 1636 cm^−1^ and 3410 cm^−1^ is attributed to the hydroxyl group of a water molecule or ferric hydroxides present in the samples [51]. The bands in the region of 400–650 cm^−1^ are attributed to the Fe-O bonds in different modes i.e., stretching and vibration modes. The band at 425 cm^−1^ and 557 cm^−1^ indicates the Fe-O bonds of magnetite nanoparticles which are similar to the studies of Basavegowda et al. [52]. The metal-oxygen band at 557 cm^−1^ corresponded to intrinsic stretching vibrations of metal at the tetrahedral site, while the metal-oxygen band found at 425 cm^−1^ was assigned to octahedral-metal stretching of Fe-O [53,54,55].

Raman spectroscopy is another important tool that helps in distinguishing the different phases of iron oxides [56]. It reveals characteristic bands for the magnetite, maghemite, and hematite [57]. Moreover, Raman spectroscopy also helps in analyzing the distance between Fe-O and thus a different vibration constant leading to two distinct bands. This means that the distortion of the spinel structure during the oxidation process leads to two Fe-O distances which are present around 660 and 710 cm^−1^ in the spectrum.

The amount of magnetite and maghemite present in IONPs can be identified by a method which was demonstrated by Schwaminger et al. [56] where there is an analysis of these bands and their ratios. In Figure 3, the Raman spectrum reveals the three characteristic peaks of IONPs i.e., 273, 488, and 691 cm^−1^, which could be due to the mixed phases of magnetite and maghemite. These mixed phases here could also be due to the utilization of high energy laser beam which may oxidize the magnetite phase and may become transformed into the maghemite phase by a phenomenon called martitization [58].

### 3.2. Diffraction Light Scattering (DLS)

The Figure 4 reveals the digital image of particle size distribution of the IONPs synthesized by the sonochemical method. The DLS graph reveals, two types of particles present in the sample out of which the size of major population varies from 450–1000 nm and 1300–1600 nm for minor population. The size of particle reported in DLS having higher values as compared to the TEM, due to the fact that in DLS, hydrodynamic sphere formed around the nanoparticles due to the difference is particle charge and dispersion medium under the influence of Brownian motion, the hydrodynamic diameter includes the core and any molecule adsorbed over the surface of the nanoparticle, which is why the particle size value seems to be higher.

TEM analysis requires the sample in the dry state whereas DLS allows the sample in the solvated state where solvent molecules associated with the nanoparticle surface (forms hydration layer) while estimating particle size by TEM, this hydration layer is not present hence lower particle size values was observed [59,60]. The average particle size of the synthesized IONPs is 550 nm and the PDI value is 0.015 which is in good agreement with the size of the IONPs.

### 3.3. X-ray Diffraction (XRD)

Figure 5 exhibit the XRD pattern of IONPs synthesized by sonochemical method. The peaks of IONPs with characteristic maximum intensity at 34.5° and 36.8° indicate the magnetite phase of iron oxide [61]. Similarly, a single major intensity peak was also observed by Predescu et al. 2018, at 35.28° for magnetite in dextran coated magnetite nanoparticles.

Besides these two peaks, there is a third peak also at 63.8° which could be due to the reflection of the magnetite phase [61]. The analyzed diffraction was matched with the standard XRD peaks of Fe_3_O_4_ with JCPDS file no: 00-003-0863 and confirmed the formation of magnetite phases. While the remaining peaks were mismatched with the standard magnetite XRD peaks that could be due to the other phases of iron oxides either maghemite in the samples. The XRD data is also in agreement with the data obtained by the Yew et al. [54] and Basavegowda et al. [52].

### 3.4. Field-Emission Scanning Electron Microscopy with Electron Diffraction Spectroscopy (FESEM-EDS) (Morphological Analysis)

The FESEM micrographs of IONPs in Figure 6 are showing two different types of particles, one is spherical shaped, nanosized whose size is varying from 10 nm to 80 nm while other particles are bulk, cuboidal shaped whose size is in microns. Similar cuboidal shaped particles were also reported by Ghanbari et al. by sonochemical synthesis [62].

The spherical particles are showing high aggregation, as aggregation is the inherent tendency of the IONPs due to the magnetic properties. Moreover, as the synthesized IONPs are the uncapped, very small size, so there is high aggregation of the particles. There are few fine granulated particles deposition on the cuboidal particles.

The purity of the synthesized IONPs was confirmed by the EDS analysis. The EDS spectra in Figure 6E shows the major peaks for Fe and O only; the Fe content was (61.22%) by weight, while the O content is (38.78%) by weight, besides these there observed slight impurities in the sample which was Na and Cl, due to improper washing of the samples.

### 3.5. Transmission Electron Microscopy (TEM) (Morphological Analysis)

Figure 7A–C shows HRTEM micrographs of IONPs synthesized by the sonochemical method. While Figure 7D, shows the histogram of particles size distribution by TEM, while Figure 7E, shows HRTEM image and d-spacing of the IONPs and Figure 7F, shows scattering area electron diffraction (SAED) pattern of IONPs synthesized by sonochemical method. Figure 7A TEM micrograph at 50 nm exhibit both cuboidal and spherical shaped nanoparticles of size 9.59 nm to 15.42 nm. Figure 7B,C at 50 nm scale exhibits IONPs of size 9.49 to 70 nm which are mainly spherical in shape. The particles are showing high aggregation due to non-use of the capping agent, smaller size and inherent tendency IONPs to exhibit aggregation. As most of the IONPs are below 30 nm, so the particles may show superparamagnetic phenomenon [63]. Figure 7D exhibit the histogram of particle size distribution of the IONPs of Figure 7C, where the average particle size of the synthesized IONPs is 39.88 nm. The HRTEM image in Figure 7E at 5 nm reveals the D-spacing and lattice fringes of the synthesized IONPs. The Figure 7F reveals the SAED pattern of the synthesized IONPs, where the absence of bright spots clearly indicates the amorphous nature of the IONPs. The amorphous nature of the synthesized IONPs are also supported by the XRD as, there is no sharp crystalline peak; rather, a broad hump was present in the sample.

### 3.6. Vibrating Sample Magnetometer (VSM) (Magnetization Study)

VSM systems are used to measure the magnetic properties of materials as a function of the magnetic field, temperature, and time. The IONPs synthesized by sonochemical method was analyzed by VSM to explore the magnetic property. The magnetization versus applied magnetic field (M–H) curve of IONPs was carried out at room temperature and the graph is shown in Figure 8. The disappearance of hysteresis with a little remanence and coercivity (Hc), indicates the absence as a long-range magnetic dipole–dipole interaction among the assemblies of superparamagnetic nanoparticles [64].

Moreover, the sigmoidal shape of the loop is characteristic of nanostructures with small magnetic fields [65]. The weak magnetism could be due to the presence of a higher amount of Fe_2_O_3_ phase which is also evidenced by the XRD [66]. The saturation magnetization (Ms) of synthesized IONPs is 2.43 emu/g while the remnant magnetization (Mr) was 0.036 emu/g and the coercivity (Hc) was 4.871 G. The magnetic measurements of the IONPs synthesized by the sonochemical method showed similar results compared to the work reported by Cheng et al. [67] and Mamani et al. [64].

### 3.7. Batch Adsorption Study of Pb and Cr Heavy Metals from Fly Ash by IONPs

#### 3.7.1. Inductively Coupled Plasma Analysis of Fly Ash Digested Sample

The inductively coupled plasma-Optical emission spectroscopy analysis of 0.3 g of fly ash was digested by aqua regia, revealed the presence of heavy metals: Al, As, Co, Cr, Cu, Hg, Pb, Ni, Pb, Zn, and non-heavy metals: Fe, Mg and Mn whose concentration is given below in Table 1. Besides this, it also has Si, and several other elements which was not calibrated in the sample. These heavy metals in fly ash come from coal being a geological sample [68]. All, these heavy metals are retained in the fly ash even after combustion of coal at high temperature. Though the value of most of the heavy metals may remain the same in both coal and fly ash, but the value of volatile heavy metals like Hg, and Cd are affected. Both being volatile in nature, a major amount of Hg and Cd are eliminated from the coal at the time of combustion in the furnace. Moreover, it has been found that smaller the size of fly ash particles higher will be the concentration of heavy metals on their surface i.e., larger fly ash particles will have lesser amount of heavy metals and other elements on their surface than the small-sized fly ash particles [69,70]. In a thermal power plant, there are two types of ashes; bottom ash and fly ash [71]. Bottom ash is generally larger in size and hence there is possibility of lesser concentrations of heavy metals on their surface [72] while the fly ash is finer in size, so comparatively there will be higher concentrations of heavy metals on their surface [72]. The composition of heavy metals along with other elements present in fly ash aqueous solution is given below in Table 1.

#### 3.7.2. Batch Adsorption Study of Heavy Metal Ions

The initial concentration of Pb^2+^ was several folds higher than the Cr ions in the 20% in the fly ash solution. Further, within one hour only lead removal was up to 89.3% which was gradually increasing till 6 h at which the lead removal efficiency reached 97.6%. Thereafter, the removal efficiency of lead decreased at 12 h, but after that, an increase in the removal efficiency was observed at 18 h and 24 h whose values were 91.4%, 93.08%, and 96.13%, respectively. While for chromium also, the highest removal efficiency was observed at one hour only which was 82.8% from where its value gradually decreased till 6 h and reached to (73.88%). Thereafter, a gradual increase in the chromium removal efficiency was observed at 12 h (77.49%) and 18 h (81.52%) and finally a minor drop at 24 h (79.83%) due to desorption. From the above observation and results it can be inferred that initially as all the adsorption sites or binding sites were vacant on the IONPs, so there was maximum adsorption of the heavy metals ions, thereafter most of the adsorption sites were occupied by the Pb^2+^ and Cr ions heavy metal ions. As a result, competition occurred among the various heavy metal ions for the limited binding sites. When all the adsorption sites are occupied and there is no further scope for the accommodation of new heavy metals ions, then it is assumed that the reaction has reached equilibrium [73,74,75]. At this point there will be maximum adsorption of the heavy metal ions, thereafter desorption will start. The maximum removal efficiency of Pb^2+^ reached 12 h (97.6%) while for Cr ions it reached one hour only (82.8%), which is given below in Figure 9. The fluctuation in the value of the heavy metal is due to the continuous adsorption and desorption from the surface of IONPs which is due to the presence and absence of binding sites for both the heavy metal ions.

## 4. Conclusions

The current study has successfully shown that the fly ash heavy metals i.e., Pb and Cr can be eliminated and removed from the aqueous solutions using amorphous IONPs within a short period of time under normal conditions. The undesired level of toxic heavy metals from fly ash can be removed by using IONPs as a nanoadsorbent, so that the fly ash could be made acceptable as a raw material for various fly ash-based industries. The fly ash free from heavy metals can be used as a safe material for making biofertilizer, panels, bricks, tiles and cements. Moreover, the study also revealed that it is possible to synthesize amorphous iron oxide nanoparticles of size 9–70 nm by sonochemical methods without any special reactions within one hour only. The sonochemical method synthesizes amorphous metal oxide nanoparticles which can be transformed into crystalline form by annealing. The IONPs was highly pure as there was mainly Fe and O as confirmed by the EDS. The IONPs were mixture of cuboidal and spherical shaped particles and it has mixed magnetite and maghemite phase of iron oxides as confirmed by Raman and XRD. Furthermore, the remediation part of this work exhibited that the IONPs can be utilized as an alternate to the conventional adsorbents for the removal of metal ions (Pb^2+^ and Cr ions) from fly ash aqueous solutions maximum removal efficiency up to 97.96% and 82.8% for Pb^2+^ and Cr ions respectively at RT. with high efficiency. Adsorption was very rapid and equilibrium was obtained within one hour and 12 h only for Pb^2+^ and Cr ions, respectively. It was also shown that adsorption was highly dependent on the initial concentration of Pb^2+^ and Cr ions in the aqueous solutions. The synthesized IONPs have the potential to remove both heavy metals and non-heavy metals from the aqueous solution in a multi-component system. The fly ash aqueous solution also has several other heavy metals ions which compete for the vacant adsorption sites on the surface of IONPs along with Pb^2+^ and Cr ions. Therefore, the fly ash free from toxic heavy metals can be accepted as a raw material for the development of fly ash-based products.

## Figures and Tables

**Figure 1 nanomaterials-10-01551-f001:**
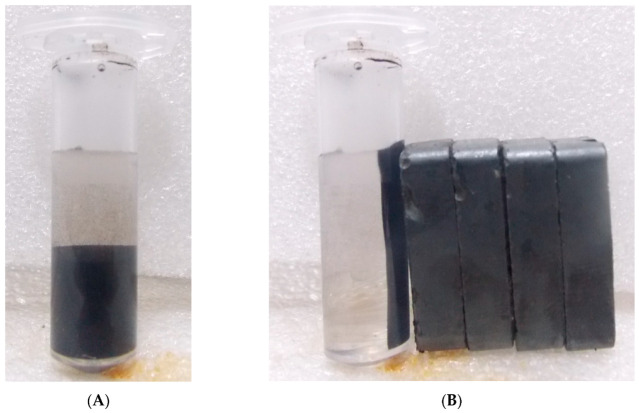
Iron oxide nanoparticles (**A**), Iron oxide nanoparticles (IONPs) responding to magnet (**B**).

**Figure 2 nanomaterials-10-01551-f002:**
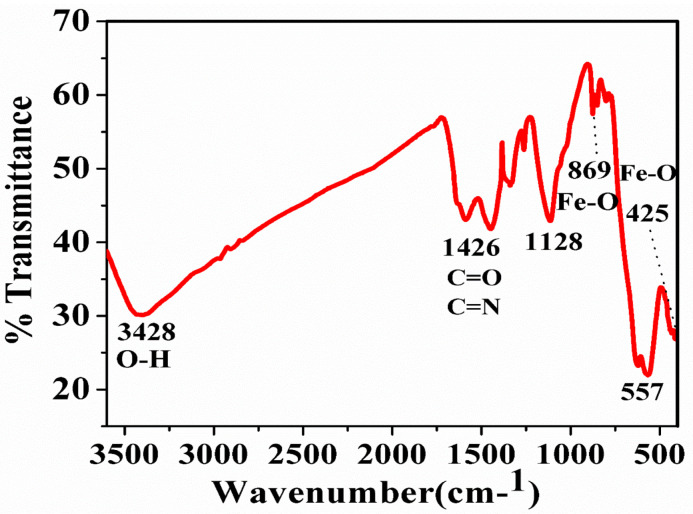
Fourier transform infrared (FTIR) spectra of IONPs synthesized by sonochemical method.

**Figure 3 nanomaterials-10-01551-f003:**
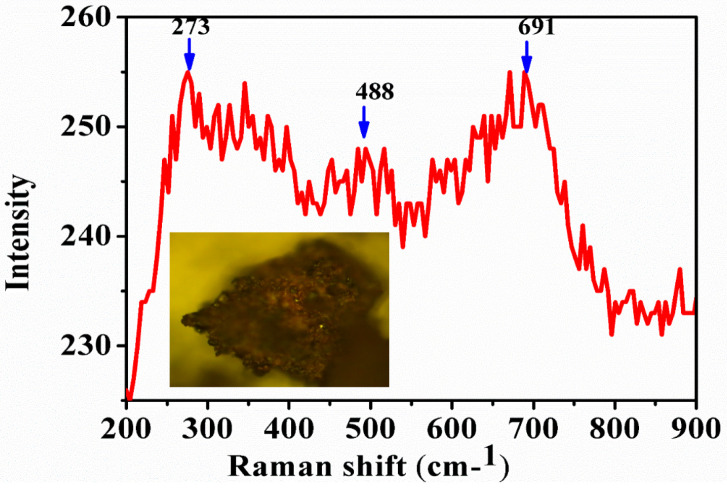
Raman spectra of IONPs synthesized by sonochemical method.

**Figure 4 nanomaterials-10-01551-f004:**
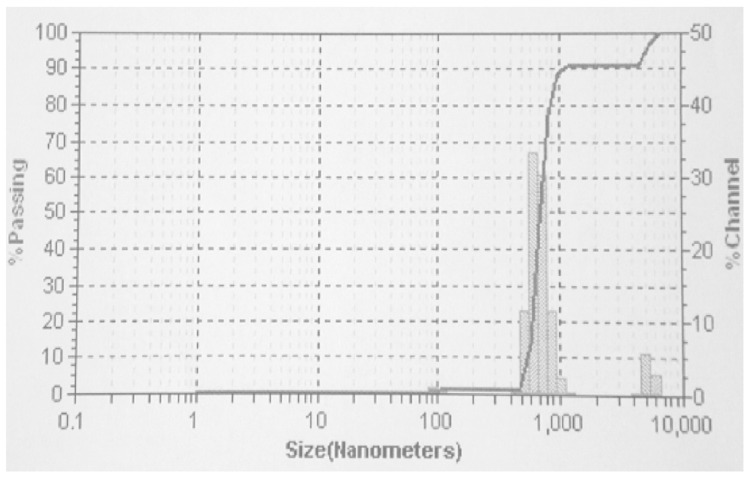
Particle size distribution of IONPs by diffraction light scattering (DLS, digital image).

**Figure 5 nanomaterials-10-01551-f005:**
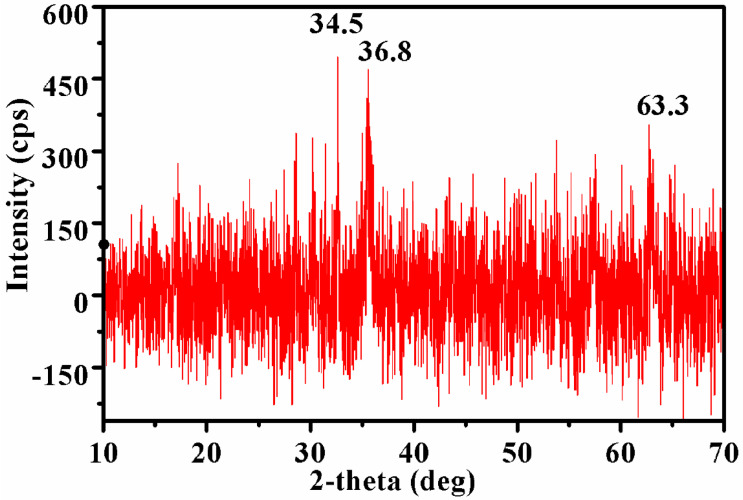
X-ray diffraction (XRD) pattern of IONPs synthesized by sonochemical method.

**Figure 6 nanomaterials-10-01551-f006:**
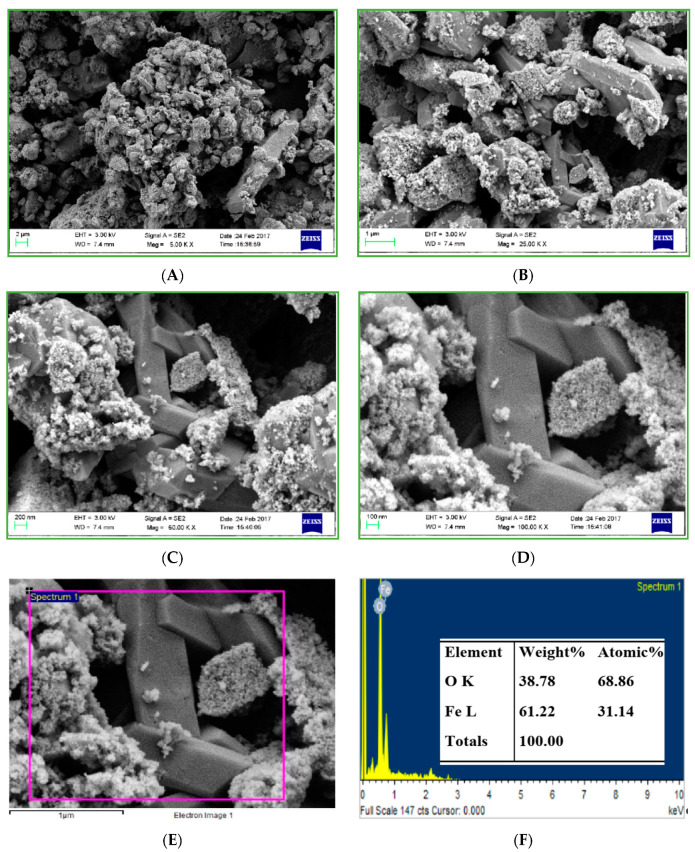
Field-emission scanning electron microscopy (FESEM) images of IONPs (**A**–**D**), electron diffraction spectroscopy (EDS) spot (**E**) and EDS spectra (**F**).

**Figure 7 nanomaterials-10-01551-f007:**
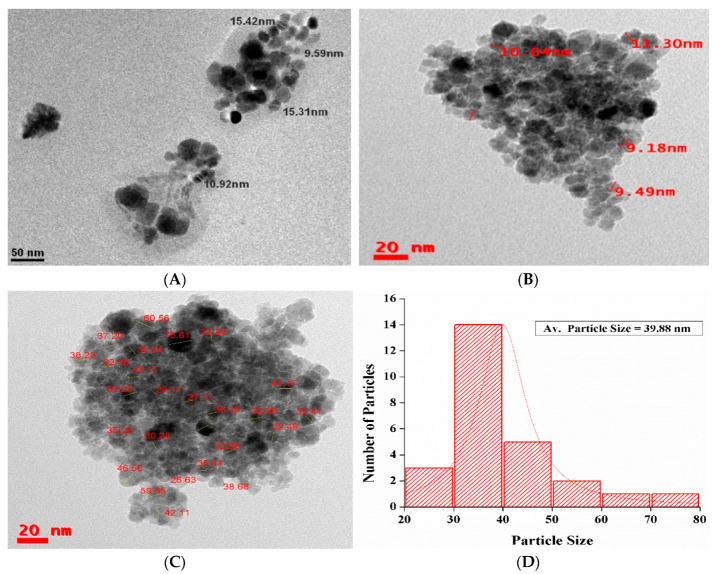
Transmission electron microscopy (TEM) and high-resolution TEM (HRTEM) micrographs of synthesized IONPs (**A**–**C**), histogram (**D**), D-spacing and lattice fringes (**E**) and SAED pattern (**F**).

**Figure 8 nanomaterials-10-01551-f008:**
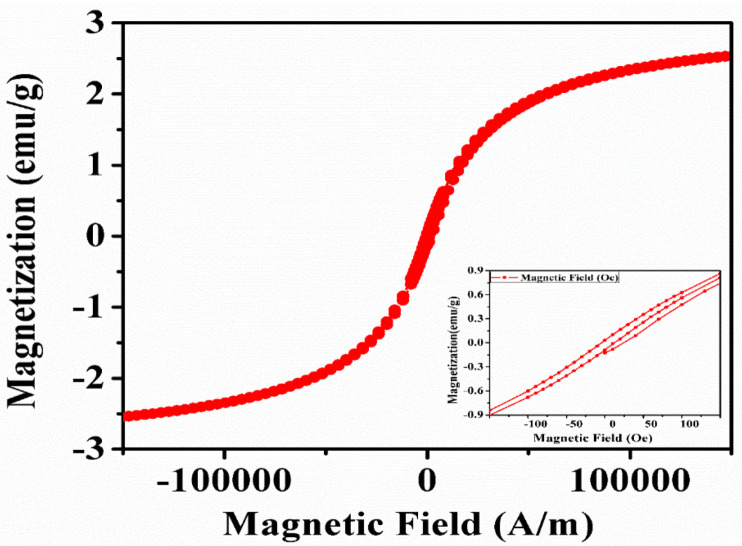
Magnetization property of IONPs against applied magnetic field.

**Figure 9 nanomaterials-10-01551-f009:**
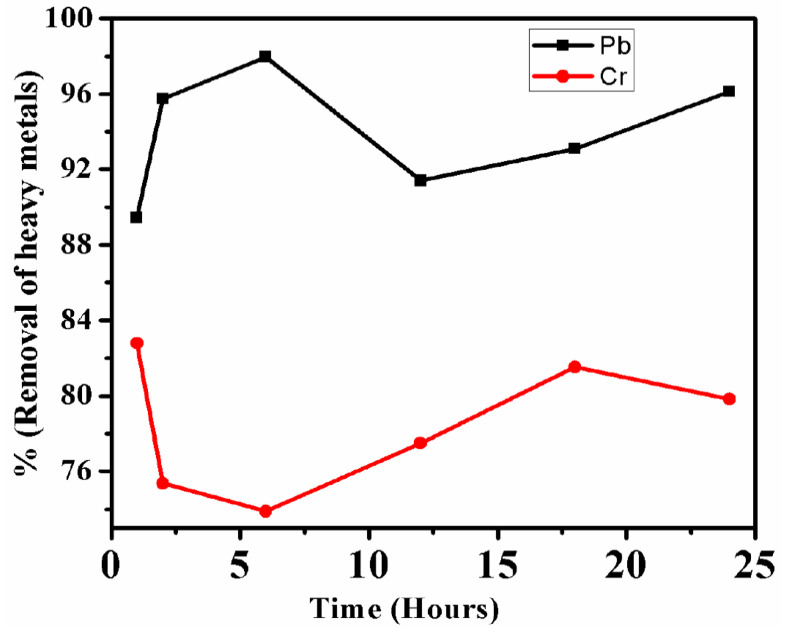
Pb and Cd removal from fly ash aqueous solution by IONPs.

**Table 1 nanomaterials-10-01551-t001:** Elements detected by ICP-OES in fly ash aqueous solution.

µg/L		mg/L										
As	Hg	Al	Cd	Co	Cr	Cu	Fe	Mg	Mn	Ni	Pb	Zn
1134.5	0.2	125.5	80.2	96.9	95.3	88.6	123.0	41.9	97.1	95.2	0.9	109.0

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
