# Peer review of "Synthesis and Characterization of Amorphous Iron Oxide Nanoparticles by the Sonochemical Method and Their Application for the Remediation of Heavy Metals from Wastewater"

_nanomaterials, 2020, doi:10.3390/nano10081551_

Round 1
Reviewer 1 Report
Paper can be accepted after the following corrections:
- Materials used for the synthesis have to be described in more details.
- SI units should be used in all paper (e.g. A/m instead of Oe, etc.)
- Quality of figures 2, 3 and 7 is not acceptable. Please correct.
- It should be verified if figure 7 presents the magnetic hysteresis loop or magnetization curve.
- In all figures: please carefully check the quality of a), b) c), etc labels.
- All abbreviations should be clearly explained.
- Conclusion should be developed, clearly presenting the most important advances presented in the paper.
- References should be numbered and clearly cited in the text of manuscript.
Author Response
Reviewer 1 1. Materials used for the synthesis have to be described in more details. Author response: Thank you for this valuable comments and suggestion. We have described all the materials which we have used in the synthesis, thoroughly in the revised manuscript as suggested. We have provided procurement details, highlighted in blue colour from line 150-156 under section 2.1 Materials. 2. SI units should be used in all paper (e.g. A/m instead of Oe, etc.) Author response: Thank you for this valuable suggestion and comment. In most of the papers the unit is Oe only while few have used kOe, so we have also used over here Oe in VSM. Besides this, all the units were converted into SI units in the revised manuscript as suggested. Here we have provided link for reference. https://www.researchgate.net/publication/274841046_Synthesis_and_Characterization_of_Fe3O4_Nanoparticles_with_Perspectives_in_Biomedical_Applications 3. Quality of figures 2, 3 and 7 is not acceptable. Please correct. Author response: The quality of all the three figures 2 (now 3), and 7 (now 8) improved in the revised manuscript as suggested. But for figure 3 (now 4), it is not possible since we were given hardcopy of the image which we have scanned and used. We don’t have raw, excel data. Thank you for this valuable comment and suggestion. 4. It should be verified if figure 7 presents the magnetic hysteresis loop or magnetization curve. Author response: Thank you for this valuable comment and suggestion. Actually fig. 7 (now 8) is showing magnetization curve, which we have corrected in the revised manuscript. We have also improved the presentation and quality of the graph. 5. In all figures: please carefully check the quality of a), b) c), etc labels. Author response: The quality of all such labels checked properly and incorporated as per the suggestion in the revised manuscript. Thank you for this comment and suggestion. 6. All abbreviations should be clearly explained. Author response: Thank you for this comment and suggestion. All the abbreviations clearly explained in the revised manuscript as suggested separately in the abstract and main text i.e. introduction in line 138 to 141. 7. Conclusion should be developed, clearly presenting the most important advances presented in the paper. Author response: Conclusion is rewritten and tried to present the most important advances presented in the paper. Changes made wherever required from line no 452-464. The authors are thankful to the reviewer for this valuable comment. 8. References should be numbered and clearly cited in the text of manuscript. Author response: All the references prepared as per the suggestion of the reviewer in the revised manuscript. All the references are numbered in the revised manuscript. Some new references are also added in the revised manuscript. Thank you for this comment and suggestion.

Reviewer 2 Report
In this paper, iron oxide nanoparticles are synthesised using a sonochemical method and then employed to adsorb lead and chromium ions. While the paper is well presented and written, there are several typographical errors and some inconsistencies in presentation (e.g., Cr(VI) and Cr6+ - use one style only). In addition, the following listed items should be addressed by the authors.
The authors need to describe clearly how this paper differs from all the literature on iron oxide nanoparticles (including numerous papers on sonochemical synthesis) and their application in the removal of heavy metal ions.
On Page 7, Line 227, the authors conclude ‘the absence of impurities in the synthesised sample’. However, this is done using EDX, which is not a very sensitive detection technique. Moreover, there appears to be a small peak at about 2.1 keV which is not assigned in Fig. 5f. The authors should revise this statement, as there may be small amounts of impurities in the sample.
On page 6, the DLS gives particles in the range 450 to 1000 nm, yet the authors conclude that the IONPs have sizes between 9 and 50 nm from TEM measurements. This requires more discussion. It would also be useful to have some indication of the variation in the sizes (eg histogram). So do the particles range in size between 9 and 1000 nm, or are the larger sizes agglomerated particles?
On page 9, details on the pH of the solution should be provided as this will influence the stability of the IONPs. Details on the other ions present in the fly ash solution should be provided.
Author Response
While the paper is well presented and written, there are several typographical errors and some inconsistencies in presentation (e.g., Cr(VI) and Cr6+ - use one style only).
A: We have changed it from Cr(VI) to Cr6+in the entire manuscript.
In addition, the following listed items should be addressed by the authors.
- The authors need to describe clearly how this paper differs from all the literature on iron oxide nanoparticles (including numerous papers on sonochemical synthesis) and their application in the removal of heavy metal ions.
Author response: The present research paper is novel and different from existing literature in terms of following terms; i) Vijay Kumar et al., reported IONPs synthesis by sonochemical method, by using inert condition, i.e. argon and time for synthesis was three hours. Ii
- ii) Roshan et al., synthesized amorphous IONPs by sonochemical method, but later on transformed into crystalline phase by annealing at high temperature. iii) Thirdly, we have suggested a method for synthesis of amorphous of IONPs, which was not transformed into crystalline phase as using high temperature will make them energy intensive step and hence expensive step. iv) We have suggested utilization of a recyclable nanoparticle for the elimination of toxic heavy metals in fly ash based industries from the raw materials. As, heavy metal loaded fly ash may not be accepted by the industries and by customers. We are trying to convert the hazardous waste into a non-hazardous waste material.
The incorporation made in the revised manuscript from line no 114 to 133.
- On Page 7, Line 227, the authors conclude ‘the absence of impurities in the synthesised sample’. However, this is done using EDX, which is not a very sensitive detection technique. Moreover, there appears to be a small peak at about 2.1 keV which is not assigned in Fig. 5f. The authors should revise this statement, as there may be small amounts of impurities in the sample.
Author response: Thank you for this valuable comment and suggestion. The statement is changed as per the suggestion in the revised manuscript, given in line no 337-339. That small peak is due to the Na formed in the reaction and remained due to improper washing of the sample.
- On page 6, the DLS gives particles in the range 450 to 1000 nm, yet the authors conclude that the IONPs have sizes between 9 and 50 nm from TEM measurements. This requires more discussion. It would also be useful to have some indication of the variation in the sizes (eg histogram). So do the particles range in size between 9 and 1000 nm, or are the larger sizes agglomerated particles?
Author response: The clarification is now added to the DLS part. As DLS and TEM shows different particles sizes due to some reason which we have now added for better understanding.
Generally, DLS provides hydrodynamic radius, there is hydration around nanoparticles as the measurement is done in the liquid sample. While in FESEM and TEM solid powder samples are used, no hydrodynamic radius and hydration layer around the nanoparticles, hence provides actual, and small size. The incorporation is made in the revised manuscript from line no 286-297.
A histogram for nanoparticle by TEM has been provided in the Figure 7 D where the average size is 38.88 nm. Yes, the larger particles are aggregated particles, as IONPs are without capping agent, and due to small size high tendency of agglomeration. Thank you for this valuable comment.
- On page 9, details on the pH of the solution should be provided as this will influence the stability of the IONPs. Details on the other ions present in the fly ash solution should be provided.
Author response: The details are now provided in the revised manuscript from line no 233-238 under desired section.
A detailed separate section added in the revised manuscript from line no 400-417 along with a table 1, mentioning all the elements present in the fly ash solution. Thank you for this valuable comment and suggestion.

Round 2
Reviewer 1 Report
Paper was corrected. However, two issues still have to be addressed:
- Please at least remove dark gray background in figure 4 using image processing software (e.g. open-source GIMP)
- I accept the use of cgs units in 3.6 chapter due to tradition. However, SI units values should be provided in brackets. Especially for values given in Oe, emu/g, G, etc. SI values should be calculated.
After these corrections paper can be accepted for publication.
Author Response
- Please at least remove dark gray background in figure 4 using image processing software (e.g. open-source GIMP)
Author response: Thank you for your suggestion and comment. The dark gray background in Fig. 4 removed up to maximum possible by using the above-mentioned software, as suggested by the reviewer in the revised manuscript.
- I accept the use of cgs units in 3.6 chapter due to tradition. However, SI units values should be provided in brackets. Especially for values given in Oe, emu/g, G, etc. SI values should be calculated.
Author response: In Fig. 8 VSM, the required and suggested cgs unit Oe has been converted into A/m i.e. SI unit as suggested by the reviewer in the revised manuscript. Thank you for the suggestion and comment.
